# MUOT-CLIP: Enhancing Few-Shot Adaptation of CLIP via Inter- and Intra- Modality Unbalanced Optimal Transport

## Abstract

Contrastive Language-Image Pre-training (CLIP) has demonstrated remarkable zero-shot capabilities across a variety of domains. To enhance its performance in data-scarce settings, few-shot adaptation methods have been developed. Other than fine-tuning the parameters (e.g., the adapter-based approach), prompt learning methods learn proper prompts to minimize the distance between the visual feature and the textual feature. Optimal Transport (OT) has proven highly effective as a measurement metric for evaluating the feature space of CLIP. However, classical OT, which forces equality constraints on both the source and target weights of the transport plan, is susceptible to noises (e.g., the misleading local regions in images and unrelated words in prompts). Furthermore, both the adapter-based and prompt learning methods usually overlook the modality gap existing in the feature space and thus risk to obtain suboptimal performance. In this paper, we extend the formulation of classical OT to unbalanced optimal transport (UOT) for better measurement. The UOT based distance measure can filter out noises adaptively. To boost the few-shot adaptation performance, a framework that measures both the inter- and intra- **M**odality distance based on **UOT** for **CLIP** is proposed, which is termed **MUOT-CLIP**. In addition, a scalable UOT solver with entropy regularization term is used for the efficient optimization of the model. Compared with the state-of-the-art methods, MUOT-CLIP consistently exhibits favorable performance on the few-shot classification benchmark of 11 datasets.

## 1 Introduction

Pretrained on large-scale web data, contrastive language-image pretraining (CLIP) (Radford et al., 2021) has witnessed widespread application in various downstream tasks (Ma et al., 2025; Wang et al., 2024; Yu et al., 2024a; Singha et al., 2023). Despite the impressive performance demonstrated by the zero-shot classification capability of CLIP (Song et al., 2022), its few-shot adaptation to novel unknown datasets remains a challenge. Many efforts have been made to further improve the performance of CLIP in data-limited scenarios (Huang et al., 2024; Zhu et al., 2024).

A typical approach is the adapter-based fine-tuning, which optimizes the parameters of the adapters added to either the image encoder or the text encoder or other selected layers with task-specific learning objective (Gondal et al., 2024; Gao et al., 2024; Zhang et al., 2022). Other than fine-tuning the parameters, typical prompt learning methods (e.g., CoOp (Zhou et al., 2022b) and CoCoOp (Zhou et al., 2022a)) learn more suitable prompts by replacing the fixed template with learnable vectors, which are optimized to minimize the distance between the visual feature and the textual feature. However, a single sentence is intuitively not sufficient to represent a class. To this end, PLOT (Chen et al., 2023) leverages optimal transport (OT) to tackle the problem that multiple prompts tend to converge to a single point when matching each prompt with the visual features respectively. However, the inherent defects make it susceptible to the irrelevant or even misleading elements of images or prompts, as shown in Figure 1(a).

Further, both the adapter-based and prompt learning methods above often overlook the modality gap existing inherent in the multimodal embeddings they have learned. Recent literature (Eslami & de Melo, 2024; Zhang et al., 2024b; Liang et al., 2022) reveals that there is a modality gap existing

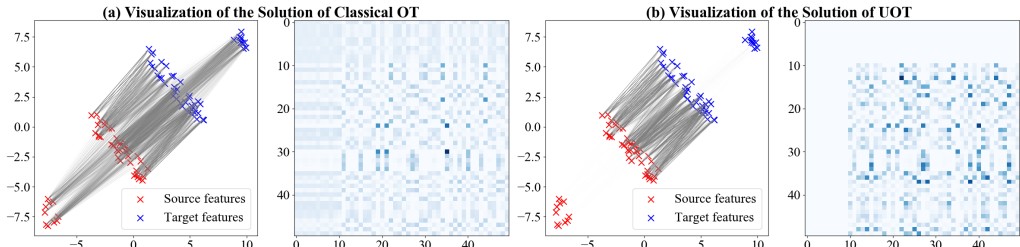

Figure 1: Visualization of the solutions of classical OT (a) and UOT (b). For each subgraph, on the left is the 2D visualization of the source samples and target samples, and on the right is the heatmap of the obtained optimal transport plan $\pi$. For more intuitive observation, we link the corresponding samples by gray lines with gray values proportional to the elements of $\pi$.

in the feature space of VLMs such as CLIP, caused by a combination of model initialization and contrastive learning optimization. The modality gap of CLIP may lead to the suboptimal classification performance under few-shot setting.

To boost the few-shot performance of CLIP, researches make improvements from two perspectives, i.e., (1) better alignment via added adapter or projector (2) taking advantage of the visual features of few-shot images. For example, SPP (Zhu et al., 2024) projects the visual and textual features into their respective subspaces to achieve alignment via local visual features. Tip-Adapter (Zhang et al., 2022) constructs the adapter using a key-value cache model from the few-shot training set.

In this paper, we analyze the fundamental flaws of classical OT that it cannot filter irrelevant or even misleading elements and propose to measure the inter- and intra- **M**odality distance via **U**nbalanced **O**timal **T**ransport for few-shot **CLIP**. A novel prompt learning based framework **MUOT-CLIP** is constructed that takes advantage of both the textual features of the learned prompts and the visual features of the retrieved prototype images, to improve the few-shot adaptation performance of CLIP on the image classification task.

Specially, in the training phase, the multiple prompts are optimized under the guidance of UOT based inter-modality distance. The barycenter of the retrieved images in the feature space is estimated to construct the visual prototype for each class. For few-shot inference, the classification of query image is predicted with both inter- and intra- modality UOT as the distance measures. In addition, a Sinkhorn-like solver is used to optimize the UOT problem in a scalable way. The main contributions of this paper can be summarized as follows,

- To achieve better measurement in the feature space, we analyze the inherent defects of classical OT and propose to measure both the inter- and intra- modality distance via extended UOT, which can adaptively filter noise.

- To mitigate the negative effect caused by modality gap and boost the few-shot adaptation performance of CLIP, the MUOT-CLIP framework that leverages both the textual features of prompts and the visual features of local retrieved images is constructed. For the computational efficiency of the overall framework, a scalable UOT solver is proposed.

- We evaluate MUOT-CLIP on 11 widely-adopted datasets for CLIP based few-shot classification. Extensive ablation studies and analysis are also conducted to validate the effectiveness of each components and explore their properties.

## 2 RELATED WORK

### 2.1 FEW-SHOT ADAPTATION OF CLIP

Few-shot adaptation focuses on improving CLIP with limited labeled data, which mainly has two types of approaches, i.e., adapter tuning and prompt learning.

**Adapter Tuning.** Adapter tuning (Liu et al., 2025; Zhang et al., 2024a; He et al., 2021) is a common method used to adapt pretrained models to downstream tasks. Adapter-based methods for CLIP

usually add adapters after the frozen image encoder or text encoder. For instance, CLIP-Adapter (Gao et al., 2024) learns new features with an additional bottleneck layer and performs residual-style feature blending with the pretrained features. Tip-Adapter (Zhang et al., 2022) makes use of the few-shot training images and prompts to construct a key-value cache model and updates the encoding prior knowledge. SPP (Zhu et al., 2024) incorporates local image features and treats them as bridges for better image-text alignment.

**Prompt Learning.** As a parameter-efficient approach, prompt learning attracts a lot of attention in the research of few-shot CLIP. CoOp (Zhou et al., 2022b) proposes to optimize unified and class-specific prompts through back-propagation, following which there are many variants. CoCoOp (Zhou et al., 2022a) addresses the generalization issue through conditional prompt learning. To learn more diverse and comprehensive prompts, PLOT (Chen et al., 2023) introduces the theory of OT to measure the distance between the textual features and the visual features. However, the formulation of OT used in (Chen et al., 2023) is limited as classical OT, which has the inherent drawback for distance measure as analyzed in Methodology. To move foreword, we extend its formulation to UOT to adaptively filter out noises without the cost of computational efficiency.

**Other Paradigms.** In addition, there are few-shot adaptation approaches adopting other paradigms, e.g., linear probing. LP++ (Huang et al., 2024) proposes a stronger baseline with specific modeling of the classifier weights, blending visual prototypes and text embeddings. CLAP (Silva-Rodriguez et al., 2024) introduces a class-adaptive linear probe objective and optimizes the balancing term via an adaptation of the general augmented Lagrangian method.

In this paper, we focus on the improvement of prompt learning based framework and boost the few-shot classification accuracy via inference-phase strategy.

## 2.2 Optimal Transport

OT quantifies the discrepancy or distance between two distributions by calculating the minimum transport cost. As a mathematical tool, recent advances have shown promising potential of OT for machine learning tasks, e.g., generative models (Zheng et al., 2024), representation learning (Yu et al., 2024b) and reinforcement learning (Asadulaev et al., 2024). The most common form is the entropy regularized classical OT solved by Sinkhorn (Cuturi, 2013), which is known for its computational efficiency. However, directly applying classical OT to few-shot CLIP may bring suboptimal performance. Tailored for the distance measure in the feature space of CLIP, we adopt the UOT formulation with scalable solver to measure both inter- and intra- modality distance.

## 3 Methodology

This section begins by illustrating the problem definition and the describe the overview of MUOT-CLIP. The property of UOT is analyzed and then the components of the model are presented in detail as well as the optimization process.

### 3.1 Problem Definition

**Few-Shot Adaptation of CLIP.** Pretrained on a vast dataset of web-based image-text pairs. CLIP (Radford et al., 2021) can adapt to the classification of unseen data without supervision , which is termed Zero-Shot CLIP for clarity. Few-shot adaptation of CLIP aims at fully leveraging the limited labeled data to enhance the performance on a novel dataset. In the few-shot setting of CLIP, there are labeled data $\{(x_i^s, y_i^s)\}$ that make up the training set $\mathbf{S}$, where $x_i$ is the image with label $y_i^s \in \{1, 2, \ldots, K\}$. The query set $\mathbf{Q}$ is composed of query images $x_i^q$ with labels $y_i^t$. The predicted label is denoted as $\hat{y}_i^t$.

**Overview.** As shown in Figure 2, MUOT-CLIP is a framework that formulates the measure of image-text distance $d_T$ and image-image $d_I$ distance as UOT problem for better few-shot adaptation capability. $d_T$ guides the learning of the prompts $\mathbf{V}_k = \{\mathcal{V}_k^{(n)}\}_{n=1}^N = \{\{v_{k,l}^n\}_{l=1}^L\}_{n=1}^N$, where $L$ is the length of $\mathcal{V}_k^{(n)}$ and $N$ is the number of prompts for each class. $k$ is the index of class. For the inference phase, the combination of both $d_T$ and $d_I$ models the probability that $x_i^t$ belongs to each class, i.e., $P(y_i^t = k | x_i^t, \mathbf{S}, \mathbf{V})$.

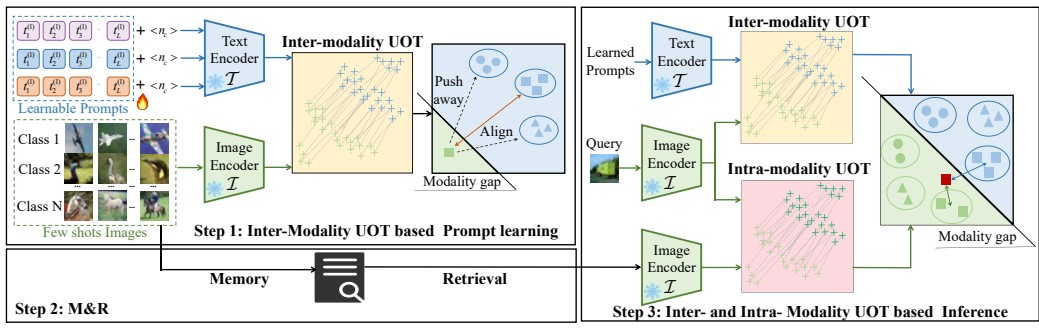

Figure 2: Overview of the architecture of the three-step MUOT-CLIP framework. In the prompt learning step, the text encoder $\mathcal{T}$ and the image encoder $\mathcal{I}$ are frozen and the learnable vectors of the prompts are optimized with $\mathcal{L}_U$ of Eq. (6). In the M&R step, the few-shot images are retrieved in a specific manner as the visual prototypes of each class. Finally, both the inter-modality distance (i.e., the distance between the feature embedding of learned prompts and query images) and the intra-modality distance (i.e., the distance between the feature embedding of retrieved prototype images and query images) are used for inference.

## 3.2 DISTANCE MEASURE IN FEATURE SPACE.

In this subsection, we analyze the motivation and the underlying principle for the UOT based distance measure. First, we revisit the definition of classical OT (Khamis et al., 2024).

**Definition 1 (Classical OT)** *Given the cost matrix $C$ (e.g., cosine similarity) from source features to target features, the classical OT is to find the optimal transport plan $\boldsymbol{\pi}$ that minimizes the total transport cost,*

$$\mathbb{W}_C := \min_{\boldsymbol{\pi} \in \Pi(\boldsymbol{\alpha}, \boldsymbol{\beta})} \langle \mathbf{C}, \boldsymbol{\pi} \rangle_F, \; \Pi(\boldsymbol{\alpha}, \boldsymbol{\beta}) := \left\{ \boldsymbol{\pi} \in \mathbb{R}_+^{M \times N} : \boldsymbol{\pi} \mathbf{1}_N = \boldsymbol{a}, \boldsymbol{\pi}^{\mathrm{T}} \mathbf{1}_M = \boldsymbol{b} \right\}, \quad (1)$$

*where $\mathbb{W}_C$ is total transport cost. $\boldsymbol{\alpha}$ and $\boldsymbol{\beta}$ are the source and target distribution respectively. $\boldsymbol{a}$ and $\boldsymbol{b}$ are the probability weights that sum up to 1.*

The key insight is that $\mathbb{W}_C$ can be viewed as the distance measure in the feature space (e.g., the distance between the visual feature and the textual feature (Chen et al., 2023)).

However, when it comes to discriminative tasks (e.g., image classification), a common case is that there are irrelevant or even misleading elements in an image, which may interfere with prediction. Moreover, due to the few-shot setting, the learnable vectors in the prompts are usually not fully tuned during the training phase. However, the definition of classical OT forces all the weights of $\boldsymbol{a}$ to be assigned to $\boldsymbol{b}$ and the constraint on the target weights must be maintained. This approach risks mistakenly correlating the important local feature with irrelevant prompts or paying too much attention to the feature of misleading regions in an image. Thus, classical OT based distance measure is not compatible with classification task to some extent and often leads to suboptimal results. In this context, we propose to extend the formulation of classical OT to UOT as follows,

**Definition 2 (UOT)** *Based on the definition of classical OT, UOT relaxes the equality constraints on both source weights and target weights via the regularization terms,*

$$\mathbb{W}_U := \min_{\boldsymbol{\pi} \in \Pi} \langle \mathbf{C}, \boldsymbol{\pi} \rangle_F + \tau_a \mathcal{D}_a(\boldsymbol{\pi} \mathbf{1}_N \| \boldsymbol{a}) + \tau_b \mathcal{D}_b(\boldsymbol{\pi}^{\mathrm{T}} \mathbf{1}_M \| \boldsymbol{b}), \; \Pi := \left\{ \boldsymbol{\pi} \in \mathbb{R}_+^{M \times N} \right\}, \quad (2)$$

*where $\mathcal{D}_a$ and $\mathcal{D}_b$ are typically set as Kullback–Leibler (KL) divergence (Van Erven & Harremos, 2014).*

The minimization objective of Eq. (2) can be rewritten as below for computational efficiency (Pham et al., 2020),

$$\hat{\mathcal{J}}_U = \langle \mathbf{C}, \boldsymbol{\pi} \rangle_F + \tau_a \mathcal{D}_a(\boldsymbol{\pi} \mathbf{1}_N \| \boldsymbol{a}) + \tau_b \mathcal{D}_b(\boldsymbol{\pi}^{\mathrm{T}} \mathbf{1}_M \| \boldsymbol{b}) - \epsilon \mathcal{H}(\boldsymbol{\pi}). \quad (3)$$

### 3.3 UOT FOR PROMPT LEARNING

We adopt the UOT based distance measure to construct the loss function and thus the learned prompts are guided to focus on more important regions of given image that are related with its class. With the text encoder $\mathcal{T}$, we can obtain the textual feature $\mathbf{G}_k = \{\mathcal{T}(\mathcal{V}_k^{(n)})\}_{n=1}^N = \{G_k^{(n)}\}_{n=1}^N \in \mathbb{R}^{N \times d}$. The visual encoder $\mathcal{I}$ has multi-head attention pooling layer whose output is $\mathbb{R}^{(H \times W + 1) \times d}$ corresponding to $M = (H \times W)$ local visual features and one global visual feature, where $H$ and $W$ are the height and width of its input feature map and $d$ is the dimension of feature embedding. With the textual feature $\mathbf{G}_k$ and visual feature $\mathbf{F}_i^s \in \mathbb{R}^{M \times d}$, we can measure the inter-modality distance between given image and the prompts belonging to class $k$ as,

$$d_T(\mathbf{F}_i^s, \mathbf{G}_k) = \mathbb{W}_U(\boldsymbol{\alpha}, \boldsymbol{\beta} | 1 - \frac{\mathbf{F}_i^s \mathbf{G}_k^T}{\|\mathbf{F}_i^s\| \|\mathbf{G}_k\|}) \in \mathbb{R}. \tag{4}$$

Then the predicted prediction probability is computed as,

$$\hat{P}(y_i^s = k | x_i^s, \mathbf{V}) = \frac{\exp((1 - d(\mathbf{F}_i^s, \mathbf{G}_k))/\tau)}{\sum_{k'=1}^K \exp((1 - d(\mathbf{F}_i^s, \mathbf{G}_{k'}))/\tau)}, \tag{5}$$

and the learning objective for the optimization of learnable vectors $v_l^n$ is,

$$\mathcal{L}_U = -\frac{1}{K} \sum_{k=1}^K \dot{y}_{i,k}^s \hat{P}(y_i^s = k | x_i^s, \mathbf{V}) \tag{6}$$

where $\dot{y}_i^s = \{\dot{y}_{i,k}\}_{k=1}^K$ denotes the one-hot encoding $y_i^s$.

### 3.4 INTER- AND INTRA- MODALITY UOT BASED INFERENCE

To obtain the prototype images for each class, a memory and retrieval (M&R) mechanism is established. Specially, the prototype images set $\hat{\mathbf{S}}$ is retrieved as subset of the set $\mathbf{S}$, i.e., $\mathbf{R} = \mathrm{MR}(\mathbf{S}) = \{(x_i^r, y_i^r)\}$. $x_i^r$ and $y_i^r$ are the retrieved few-shot images and the corresponding labels respectively. There are different implementations for image retrieval, e.g., full retrieval or $d(\mathbf{F}_i^s, \mathbf{G}_k^*)$ guided partial retrieval, where $\mathbf{G}_k^*$ is the learned prompt. We estimate the barycenter of class $k$ retrieved images in the feature space as,

$$\bar{\mathbf{F}}_k^s = \frac{1}{H} \sum_{i=1}^H \mathcal{I}(x_i^r) \in \mathbb{R}^{M \times d} \tag{7}$$

To mitigate the modality gap between the visual feature and the textual feature, we propose to measure the probability $P(\hat{y}_i^t = k | x_i^t, \mathbf{S}, \mathbf{V})$ from the perspective of both inter- and intra- modality distance. The inter-modality distance can be calculated based on the formulation of Eq. (4), Taking into account both text-image and image-image similarity, the probability $P(\hat{y}_i^t = k | x_i^t, \mathbf{S}, \mathbf{V})$ can be measured indirectly via,

$$
\begin{aligned}
d_k &= \mu d_T(\mathbf{F}_i^q, \mathbf{G}_k^*) + (1 - \mu) d_I(\mathbf{F}_i^q, \bar{\mathbf{F}}_k^s) \\
&= \mathbb{W}_U(\boldsymbol{\alpha}, \boldsymbol{\beta} | 1 - \frac{\mathbf{F}_i^q \mathbf{G}_k^{*T}}{\|\mathbf{F}_i^q\| \|\mathbf{G}_k^*\|}) + \mathbb{W}_U(\boldsymbol{\alpha}, \boldsymbol{\beta} | 1 - \frac{\mathbf{F}_i^q \bar{\mathbf{F}}_k^{sT}}{\|\mathbf{F}_i^q\| \|\bar{\mathbf{F}}_k^s\|}),
\end{aligned} \tag{8}
$$

where $\mu \in [0, 1]$ is the weighted factor. The class $k$ with the minimum $d_k$ is the predicted class $\hat{y}_i^t$.

### 3.5 MODEL OPTIMIZATION

MUOT-CLIP is basically a three-step framework that performs prompt learning guided by $d_T$, estimates the barycenter of the retrieved images belonging to each class $k$ in the feature space, and then predicts the classification of the query image with $d_T$ and $d_I$. The component that counts for the efficiency and performance of MUOT-CLIP is the UOT module. For computational efficiency and GPU compatibility, we use the following solver to optimize the UOT problem in a scalable way. For

the UOT problem with the minimization objective formulation of Eq. (3), $\mathcal{D}_a$ and $\mathcal{D}_b$ are set as KL divergence. We tackle it in an iterative manner,

$$
\begin{cases}
f_i^{'(t)} = \left[ \dfrac{a_i}{\sum_{j=1}^N g_j^{'(t-1)} \exp\left(-\frac{C_{ij}}{\epsilon}\right)} \right]^{\frac{\tau_a}{\tau_a+\epsilon}} \\[4mm]
g_j^{'(t)} = \left[ \dfrac{b_j}{\sum_{i=1}^M f_i^{'(t)} \exp\left(-\frac{C_{ij}}{\epsilon}\right)} \right]^{\frac{\tau_b}{\tau_b+\epsilon}}
\end{cases},
\tag{9}
$$

where $f_i^{'} = \exp(f_i/\epsilon)$, $g_j^{'} = \exp(g_j/\epsilon)$ and $\boldsymbol{f}$, $\boldsymbol{g}$ are the dual variables of UOT. $t$ is the current number of iterations. The computed transport plan of the $t$th iteration is,

$$
\pi_{ij}^{(t)} = f_i^{'(t)} g_j^{'(t)} \exp\left(-\frac{C_{ij}}{\epsilon}\right)
\tag{10}
$$

Then the UOT based distance is calculated as, $\mathbb{W}_U = \langle \mathbf{C}, \boldsymbol{\pi}^* \rangle_F$, where $\pi^*$ is the optimal transport plan. Please refer to the appendix in the supplementary material for detailed proof.

# 4 EXPERIMENTS

## 4.1 EXPERIMENTS SETUP

**Datasets.** We conduct few-shot classification experiments on 11 datasets, including OxfordPets (Parkhi et al., 2012), Flowers102 (Nilsback & Zisserman, 2008), FGVCAircraft (Maji et al., 2013) , DTD (Cimpoi et al., 2014), EuroSAT (Helber et al., 2019), StanfordCars (Krause et al., 2013), Food101 (Bossard et al., 2014), SUN397 (Xiao et al., 2010), Caltech101 (Fei-Fei et al., 2004), UCF101 (Soomro et al., 2012), and ImageNet (Deng et al., 2009).

**Baselines.** We compare MUOT-CLIP with three categories of approaches, (1) Adapter-based methods: Tip-Adapter (Zhang et al., 2022), CLIP-Adapter (Gao et al., 2024), and SSP (Zhu et al., 2024) (2) prompt learning methods: CoOp (Zhou et al., 2022b) and PLOT (Chen et al., 2023) (3) LP++ (Huang et al., 2024), a linear probing method that achieves performance comparable to other baselines. For effective comparison, we choose the fine-tuned version of Tip-Adapter (Zhang et al., 2022), which is termed Tip-Adapter-F.

**Implementation Details.** MUOT-CLIP is constructed on CoOp (Zhou et al., 2022b), with the position of the class token at the end and the random parameter initialization strategy. The length of learnable context tokens is set as 16. Unless otherwise stated, the number of prompts is $N = 4$. Following previous work (Zhou et al., 2022b; Chen et al., 2023), we adopt ResNet-50 (He et al., 2016) as the backbone of the image encoder. The number of local visual feature is $M = 7 \times 7$. We train MUOT-CLIP via the SGD optimizer with 0.002 initial learning rate, CosineAnnealingLR schedule, and a warmup trick with $10^{-5}$ learning rate. For small datasets such as FGVCAircraft (Maji et al., 2013), Flowers102 (Nilsback & Zisserman, 2008), and StanfordCars (Krause et al., 2013), the batch size is set as 32, while for the larger datasets such as ImageNet (Deng et al., 2009) and SUN397 (Xiao et al., 2010), the batch size is set as 128. All experiments are executed on 4 NVIDIA RTX 4090D GPUs. Please refer to the appendix in the supplementary material for more details.

## 4.2 MAIN RESULTS

**Overall Performance.** In Table 1, we report the quantitative results obtained by MUOT-CLIP and the recent literature in the few-shot image classification task. we present the accuracy (i.e., the percentage of correct predictions) on each dataset and the average accuracy over all the 11 datasets, with 1,2,4,8,16 shots images for training respectively. For the accuracy of each method, we report the average of 3 runs. The first observation is that MUOT-CLIP achieves the best average accuracy on all the shots settings. For the accuracy on each dataset with different shots settings, MUOT-CLIP tops more than half of the rankings. Secondly, we can observe the increase of the accuracy with more shot images, which shows that MUOT-CLIP benefits from more training data while still achieves comparable performance under the extreme 1-shot setting. In addition, the performance advantage of MUOT-CLIP over the baselines does not come at the cost of significantly more training latency

Table 1: The classification accuracy (%) comparison on few-shot image classification tasks across 11 datasets.The datasets include Pets.(OxfordPets), F102.(Flowers102), FGVC.(FGCVAircraft), DTD, Euro(EuroSAT), Cars.(StanfordCars), F101.(Food101), SUN.(SUN397), C101.(Caltech101), UCF.(UFC101), and ImgN.(ImageNet). To compare the overall performance of each method, the average accuracy (%) across all the 11 datasets with the number of shots fixed is reported. The bold value indicates the optimal under its experimental setup, whereas the underlined indicates the suboptimal.We report the average of accuracy of 3 runs. Please refer to the supplementary material for the variance and more detailed data.

| Dataset | Pets. | F102. | FGVC. | DTD | Euro. | Cars. | F101. | SUN. | C101. | UCF. | ImgN. | Avg |
|---|---|---|---|---|---|---|---|---|---|---|---|---|
| Zero-shot CLIP | 85.77 | 66.14 | 17.28 | 42.32 | 37.56 | 55.61 | 77.31 | 58.52 | 86.29 | 61.46 | 58.18 | 58.77 |
| | | | | | 1 shot | | | | | | | |
| CoOp | 85.89 | 68.12 | 9.64 | 44.39 | 50.63 | 55.59 | 74.32 | 60.29 | 87.53 | 61.92 | 57.15 | 59.59 |
| Tip-Adapter-F | 85.70 | 67.73 | 18.23 | 46.92 | 47.63 | 57.24 | 77.53 | 61.02 | 87.35 | 64.28 | 60.59 | 61.29 |
| PLOT | **87.27** | 72.00 | 17.77 | 47.23 | 56.20 | 56.17 | **78.03** | 62.63 | 89.03 | 64.37 | 57.90 | 62.60 |
| CLIP-Adapter | 85.99 | 73.49 | 17.49 | 45.80 | 61.40 | 55.13 | 76.82 | 61.30 | 88.60 | 62.20 | 61.20 | 62.67 |
| LP++ | 84.24 | **78.21** | 19.69 | 46.97 | 57.23 | 57.20 | 76.61 | 62.47 | 88.56 | 65.41 | 61.18 | 63.43 |
| Tip+SSP | 86.32 | 76.05 | 19.74 | 46.81 | 59.17 | 57.60 | 77.58 | 61.49 | 88.76 | 63.02 | **61.71** | 63.48 |
| Ours | 87.20 | 75.30 | **20.10** | 48.87 | 62.20 | 57.63 | 77.97 | 63.43 | 89.67 | 66.17 | 59.63 | **64.38** |
| | | | | | 2 shot | | | | | | | |
| CoOp | 82.64 | 77.51 | 18.68 | 45.15 | 61.50 | 58.28 | 72.49 | 59.48 | 87.93 | 64.09 | 57.81 | 62.32 |
| Tip-Adapter-F | 86.05 | 68.18 | 19.12 | 48.50 | 57.62 | 58.12 | 77.53 | 62.15 | 88.17 | 65.48 | 61.4 | 62.94 |
| PLOT | 87.13 | 82.00 | 19.37 | 51.17 | 64.00 | 58.40 | 77.90 | 62.37 | 89.87 | 67.83 | 59.90 | 65.45 |
| CLIP-Adapter | 86.73 | 81.61 | 20.10 | 51.48 | 63.90 | 58.74 | 77.22 | 63.29 | 89.37 | 67.12 | 61.52 | 65.55 |
| LP++ | 85.74 | **84.69** | 21.58 | 52.44 | 61.65 | **59.95** | 77.22 | **64.65** | 89.53 | 69.20 | 61.56 | 66.20 |
| Tip+SSP | 87.03 | 79.5 | **22.71** | 50.77 | 62.36 | 59.11 | 77.62 | 62.84 | 89.01 | 66.09 | **61.82** | 65.35 |
| Ours | **87.73** | 82.83 | 21.10 | **54.57** | 66.17 | 58.80 | **78.13** | 63.77 | 90.67 | 69.87 | 60.20 | **66.71** |
| | | | | | 4 shot | | | | | | | |
| CoOp | 86.70 | 86.20 | 21.87 | 53.49 | 70.18 | 62.62 | 73.33 | 63.47 | 89.55 | 67.03 | 59.99 | 66.77 |
| Tip-Adapter-F | 86.40 | 71.17 | 20.55 | 57.16 | 69.30 | 59.34 | 77.82 | 63.86 | 89.49 | 67.61 | 62.12 | 65.89 |
| PLOT | 88.57 | 88.27 | 22.80 | 55.70 | 70.77 | 62.70 | 77.30 | 65.27 | 90.67 | 70.67 | 60.37 | 68.46 |
| CLIP-Adapter | 87.46 | 87.17 | 22.59 | 56.86 | 73.38 | 62.45 | 77.92 | 65.96 | 89.98 | 69.05 | 61.84 | 68.61 |
| LP++ | 86.94 | **89.56** | 24.22 | 57.75 | 68.67 | **63.44** | 77.79 | **67.28** | 90.87 | 71.68 | 62.55 | 69.16 |
| Tip+SSP | 86.81 | 84.13 | 23.67 | 54.79 | 67.21 | 61.47 | 77.64 | 64.27 | 90.14 | 67.80 | 61.98 | 67.26 |
| Ours | **88.63** | 88.60 | **24.93** | 58.27 | 75.30 | 63.33 | 77.93 | 66.20 | 91.73 | 73.23 | 60.77 | **69.90** |
| | | | | | 8 shot | | | | | | | |
| CoOp | 85.32 | 91.18 | 26.13 | 59.97 | 76.73 | **68.43** | 71.82 | 65.52 | 90.21 | 71.94 | 61.56 | 69.89 |
| Tip-Adapter-F | 87.66 | 84.11 | 23.60 | 62.38 | 75.22 | 64.25 | 78.26 | 67.25 | 90.54 | 72.05 | 63.41 | 69.88 |
| PLOT | 87.17 | 92.63 | 26.63 | 61.60 | 78.00 | 67.03 | 75.50 | 66.43 | 90.93 | 75.80 | 60.57 | 71.12 |
| CLIP-Adapter | 87.65 | 91.72 | 26.25 | 61.00 | 77.93 | 67.89 | 78.04 | 67.50 | 91.40 | 73.30 | 62.68 | 71.40 |
| LP++ | **87.71** | 92.61 | 27.73 | 62.42 | 75.86 | 67.81 | **78.53** | 69.34 | 91.84 | 74.86 | **63.76** | 72.04 |
| Tip+SSP | 87.19 | 88.63 | 27.78 | 58.98 | 72.28 | 63.89 | 77.75 | 65.68 | 90.91 | 69.28 | 62.22 | 69.51 |
| Ours | 87.50 | **93.03** | **28.33** | 63.43 | 80.53 | 67.47 | 76.13 | 67.80 | 91.93 | 77.30 | 61.70 | **72.29** |
| | | | | | 16 shot | | | | | | | |
| CoOp | 87.01 | 94.51 | 31.26 | 63.58 | 83.53 | 73.36 | 74.67 | 69.26 | 91.83 | 75.71 | 62.95 | 73.42 |
| Tip-Adapter-F | **89.08** | 93.02 | 30.37 | 65.23 | 78.59 | 71.38 | **78.99** | 70.94 | 92.10 | 77.30 | **65.06** | 73.82 |
| PLOT | 87.10 | **95.37** | 31.30 | 65.20 | 82.70 | 72.90 | 77.17 | 69.63 | 92.93 | 77.77 | 60.67 | 73.89 |
| CLIP-Adapter | 87.84 | 93.90 | 32.10 | 65.96 | 84.43 | 74.01 | 78.25 | 69.55 | 92.49 | 76.76 | 63.59 | 74.44 |
| LP++ | 88.38 | 94.26 | 31.73 | 66.40 | 80.53 | 72.33 | 78.88 | 71.23 | 92.73 | 77.46 | 64.73 | 74.42 |
| Tip+SSP | 88.83 | 90.62 | 30.18 | 62.23 | 73.62 | 67.19 | 77.94 | 67.01 | 91.56 | 70.95 | 62.75 | 71.17 |
| Ours | 87.70 | 95.13 | **33.53** | **66.57** | 82.93 | 72.57 | 77.47 | 70.30 | **93.37** | 78.83 | 62.23 | **74.60** |

or parameter size. Please refer to the supplementary material for details of the execution time of MUOT-CLIP compared with the baseline.

**Comparison with and CoOp and PLOT.** Since the MUOT-CLIP model is constructed on the base of typical prompt learning methods i.e., CoOp (Zhou et al., 2022b) and PLOT (Chen et al., 2023), we focus on the comparison of MUOT-CLIP with both CoOp and PLOT. As shown in Table 1, MUOT-CLIP performs better than both of them on most of the 11 datasets and exhibits relatively significant advantage in terms of average accuracy. This empirically proves the effectiveness of the UOT based inter- and intra- modality distance measure.

## 4.3 ABLATION STUDY AND FURTHER ANALYSIS

**Effectiveness of Different Components.** We conduct ablations in Table 2 and report the accuracy of different versions on the DTD (Cimpoi et al., 2014) and FGVCAircraft (Maji et al., 2013) datasets with 1,2,4,8,16 shots images for training respectively. The "T-I" denotes inter-modality text-image

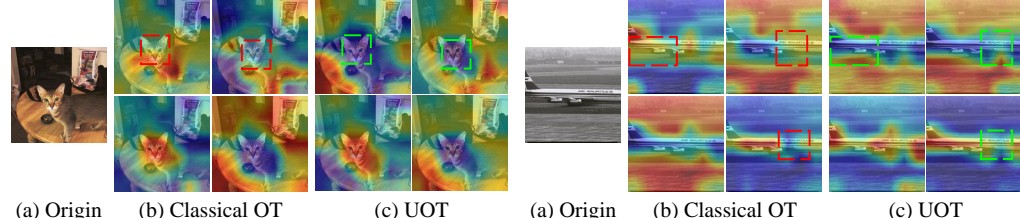

| (a) Origin | (b) Classical OT | (c) UOT | | (a) Origin | (b) Classical OT | (c) UOT |

Figure 3: Visualization of the heatmaps of the optimal transport plan $\pi$ produced by classical OT and UOT. Images of (a) are the original images. The red boxes in (b) indicate the unreasonable part of the heatmap and the corresponding regions are bounded by the green boxes in (c).

Table 2: The ablation study of UOT-based inter- and intra- modality distance on DTD and FGV-CAircraft datasets. For simplicity, the "T-I" denotes inter-modality text-image distance and the "I-I" denotes intra-modality image-image distance.

| T-I | I-I | DTD | | | | | FGVC | | | | |
|-----|-----|-----|-----|-----|-----|-----|------|-----|-----|-----|-----|
| | | 1 | 2 | 4 | 8 | 16 | 1 | 2 | 4 | 8 | 16 |
| ✗ | ✗ | 47.23 | 51.17 | 55.70 | 61.60 | 65.20 | 17.77 | 19.37 | 22.80 | 26.63 | 31.30 |
| ✓ | ✗ | 47.37 | 53.30 | 55.93 | 62.63 | 66.20 | 19.13 | 19.73 | 22.90 | 27.13 | 32.03 |
| ✗ | ✓ | 47.67 | 51.90 | 57.97 | 62.57 | 66.10 | 19.07 | 20.50 | 24.17 | 27.67 | 32.60 |
| ✓ | ✓ | **48.87** | **54.57** | **58.27** | **63.43** | **66.57** | **20.10** | **21.10** | **24.93** | **28.33** | **33.53** |

distance and the "I-I" denotes intra-modality image-image distance. The version with both "T-I" and "I-I" is the complete MUOT-CLIP model. The version without the former refers to replacing the UOT module used for measuring the inter-modality distance with the classical OT. The version without latter performs inference only based on the inter-modality distance. As the baseline in this experiment, the version without both of them degenerates to PLOT (Chen et al., 2023). The versions that adopt UOT to measure only inter- or intra- modality distance outperform the baseline under all the settings of Table 2, which proves the individual contributions of these components. In addition, the complete version of the MUOT-CLIP model achieves better accuracy than its variants. This shows the effectiveness and advantages of the framework design of MUOT-CLIP.

**Visualization of the Optimal Transport Plan.** The definition of the formulation of classical OT and UOT determines that the distance measure based on classical OT is inevitably affected by noises, whereas that based on UOT can adaptively filter out noises. For more intuitive analysis, we visualize the heatmap of the optimal transport plan $\pi$ for images of two different classes. UOT can capture the outline of the target more completely, while fragmentation and confusion are observed in the heatmap of $\pi$ produced by classical OT, as shown in the red boxes of Figure 3(b), which is the reason for the advantage of the UOT based MUOT-CLIP over PLOT.

**Performance with More Shots.** Table 1-2 conduct few-shot image classification experiments with no more than 16 shots images. We evaluate the performance change of MUOT-CLIP with more shots images. Specially, other than the common setting of 1, 2, 4, 8, 16 shots, we conduct additional experiments with 32 and 64 shots images on the DTD, UFC101, EuroSAT, and ImageNet datasets. As shown in Table 3, a significant trend is that more shots images bring higher classification accuracy. This can be factorized into two aspects:(1) more shots images improve the quality of learned prompts through prompt learning (2) more shots images lead to more effective intra-modality measure during inference because the verage visual feature of prototype images can be more representative for the corresponding class with more retrieved images. This also demonstrates the robustness of MUOT-CLIP that not only works in typical few-shot settings, but also in scenarios with more labeled data.

**Prompts Number Ablation.** In this paper, we follow the setting of PLOT (Chen et al., 2023) to set the number of prompts as 4 for fairness. To evaluate the effect of the number of prompts $N$ on the performance of the proposed method, we evaluate with different settings of $N$ in Table 4 and find that $N = 4$ is enough and more prompts cannot bring a significant improvement.

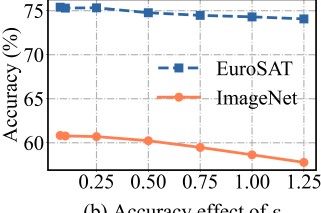 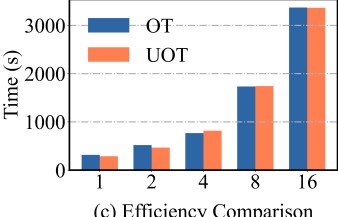

(a) Accuracy effect of $\tau_a$ and $\tau_b$     (b) Accuracy effect of $\varepsilon$     (c) Efficiency Comparison

Figure 4: Parameter sensitivity analysis of $\tau_a$, $\tau_b$ and $\epsilon$. (a) Accuracy effect of $\tau_a$ and $\tau_b$. (b) Accuracy effect of $\epsilon$. (c) Running time of PLOT and MUOT-CLIP with the change of shots.

Table 3: The analysis study to observe the change of the image classification accuracy of MUOT-CLIP with more shots on four datasets. The bold value indicates the number of shots with the best accuracy on the same dataset.

| Shots | 1 | 2 | 4 | 8 | 16 | 32 | 64 |
|---|---|---|---|---|---|---|---|
| DTD | 48.87 | 54.57 | 58.27 | 63.43 | 66.57 | 69.33 | **71.83** |
| UCF. | 66.17 | 69.87 | 73.23 | 77.30 | 78.83 | 80.23 | **81.30** |
| Euro. | 62.20 | 66.17 | 75.30 | 80.53 | 82.93 | 85.87 | **88.30** |
| ImgN. | 59.63 | 60.20 | 60.77 | 61.70 | 62.23 | 62.43 | **63.00** |

**Parameter Sensitivity Study.** Figure 4 reports the accuracy effect of $\tau_a$, $\tau_b$, and $\epsilon$. The finding is that when the values of $\tau_a$ and $\tau_b$ approach 0 in Figure 4(a), the accuracy of MUOT-CLIP drops significantly. Since UOT will degenerates into classical OT with both $\tau_a$ and $\tau_b$ approaching 0, the advantages of UOT over classical OT can be further verified. In Figure 4(b), the accuracy drops as $\epsilon$ increases, which is consistent with the fact that the larger the weight of the entropy regularization term in Eq. (3), the larger the effect of the approximation approach and the less accurate the solution.

Table 4: The analysis study to evaluate the effect of $N$. We conduct experiments on the DTD, UCF.(UFC101), Euro.(EuroSAT), and SUN.(SUN397) datasets with $N$ changes.

| # of Prompts | $N = 1$ | $N = 2$ | $N = 4$ | $N = 8$ |
|---|---|---|---|---|
| DTD | 57.77 | 58.10 | 58.27 | **58.33** |
| UCF. | 71.57 | 72.43 | **73.23** | 73.10 |
| Euro. | 74.20 | 75.20 | **75.30** | 75.17 |
| SUN. | 66.13 | 65.93 | 66.20 | **66.93** |

**Efficiency Analysis.** To evaluate the efficiency of MUOT-CLIP, we calculate the execution time of PLOT and MUOT-CLIP. Specially, we choose a relatively small dataset EuroSAT and a larger dataset ImageNet to analyze the scalability of the methods. The total execution time on EuroSAT and ImageNet are reported in Figure 4(c). Due to the scalable UOT solver with entropy regularization term, the latency of MUOT-CLIP is modest on EuroSAt compared to PLOT. On the larger ImageNet dataset, the execution time of MUOT-CLIP is less than that of PLOT in most cases.

## 5 CONCLUSIONS

In this paper, we first investigate the limitation of current prompt learning based methods for few-shot CLIP and the modality gap existing in the feature space of CLIP. Starting from the observation that classical OT cannot filter out noises when measuring the distance in feature space, we propose to extend the formulation of it to UOT. To mitigate the effect of the modality gap, we leverage UOT to measure both inter- and intra- modality distance and construct a novel framework MUOT-CLIP for the few-shot adaptation of CLIP. Without introducing additional learnable parameters other than the learnable prompts, MUOT-CLIP outperforms the recent state-of-the-art methods in the few-shot classification task. We believe that research on the measure and property of visual and textual feature embedding under few-shot scenario will promote the real-world application of VLMs.

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

# A  APPENDIX

## A.1  PROOF FOR THE SCALABLE SOLUTION OF UOT

Given any unbalanced optimal transport problem with KL-Divergence with regularization:

$$\min_{\boldsymbol{\pi} \geq 0} \hat{\mathcal{J}}_U = \langle \mathbf{C}, \boldsymbol{\pi} \rangle_F + \tau_a \mathcal{D}_a(\boldsymbol{\pi} \mathbf{1}_N \| \boldsymbol{a}) + \tau_b \mathcal{D}_b(\boldsymbol{\pi}^\mathrm{T} \mathbf{1}_M | \boldsymbol{b}) - \epsilon \mathcal{H}(\boldsymbol{\pi})$$

$$\text{s.t. ( Optional ) :} \ \boldsymbol{\pi} \mathbf{1}_N = \boldsymbol{a}, \quad \boldsymbol{\pi}^\top \mathbf{1}_M = \boldsymbol{b},$$

where $\mathcal{H}(\boldsymbol{\pi}) = -\langle \boldsymbol{\pi}, \log(\boldsymbol{\pi}) - 1 \rangle$. The constraints are optional for the following UOT deduction. The notions of $M$ and $N$ here refer to the number of source and target distribution samples, not specifically the number of local image features and the number of prompts.

The Lagrange multipliers of UOT with KL-Divergence is given as:

$$\max_{\boldsymbol{f}, \boldsymbol{g}} \min_{\boldsymbol{\pi} \geq 0} \hat{\mathcal{J}}_U = \tau_a \mathrm{KL}\left(\boldsymbol{\pi} \mathbf{1}_N \| \boldsymbol{a}\right) + \langle \boldsymbol{f}, \boldsymbol{\pi} \mathbf{1}_N \rangle + \tau_b \mathrm{KL}\left(\boldsymbol{\pi}^\top \mathbf{1}_M \| \boldsymbol{b}\right) +$$

$$\langle \boldsymbol{g}, \boldsymbol{\pi}^\top \mathbf{1}_M \rangle - \epsilon \mathcal{H}(\boldsymbol{\pi}) + \mathscr{C}_{\mathrm{EUOT}},$$

Please note that the $\boldsymbol{f}$ and $\boldsymbol{g}$ here are dual variables of EUOT. Then,

$$\mathscr{C}_{\mathrm{EUOT}} = \left\langle \boldsymbol{C} - \boldsymbol{f} \otimes \mathbf{1}_N^\top - \mathbf{1}_M \otimes \boldsymbol{g}^\top, \boldsymbol{\pi} \right\rangle = \sum_{i,j} \left(C_{ij} - f_i - g_j\right) \pi_{ij}.$$

Taking the differentiation on $\boldsymbol{\pi}_{ij}$, we can obtain the following results:

$$\frac{\partial \mathcal{J}}{\partial \pi_{ij}} = \tau_a \log \frac{\sum_{j=1}^N \pi_{ij}}{a_i} + f_i + \tau_b \log \frac{\sum_{i=1}^M \pi_{ij}}{b_j} + g_j \left[ (C_{ij} - f_i - g_j) + \epsilon \log \pi_{ij} \right]$$

$$= C_{ij} + \tau_a \log \frac{\sum_{j=1}^N \pi_{ij}}{a_i} + \tau_b \log \frac{\sum_{i=1}^M \pi_{ij}}{b_j} + \epsilon \log \pi_{ij} = 0.$$

Then we can setup as:

$$\begin{cases} \sum_{j=1}^N \pi_{ij} = a_i \exp\left(-\frac{f_i}{\tau_a}\right) \\ \sum_{i=1}^M \pi_{ij} = b_j \exp\left(-\frac{g_j}{\tau_b}\right) \end{cases}, \ \pi_{ij} = \exp\left(\frac{f_i + g_j - C_{ij}}{\epsilon}\right).$$

Therefore, we can obtain the important result as we expected.

$$\min_{\boldsymbol{f}, \boldsymbol{g}} \mathcal{J}_\mathrm{E} = \tau_a \left\langle \boldsymbol{a}, \exp\left(-\frac{\boldsymbol{f}}{\tau_a}\right) \right\rangle + \tau_b \left\langle \boldsymbol{b}, \exp\left(-\frac{\boldsymbol{g}}{\tau_b}\right) \right\rangle + \epsilon \sum_{i=1}^M \sum_{j=1}^N \exp\left(\frac{f_i + g_j - C_{ij}}{\epsilon}\right).$$

We can optimize $\mathcal{J}_\mathrm{E}$ as follows. First, we optimize $\boldsymbol{f}$ as:

$$\frac{\partial \mathcal{J}_\mathrm{E}}{\partial f_i} = -a_i \exp\left(-\frac{f_i}{\tau_a}\right) + \exp\left(\frac{f_i}{\epsilon}\right) \sum_{j=1}^N \exp\left(\frac{g_j - C_{ij}}{\epsilon}\right) = 0,$$

and thus,

$$\exp\left(\frac{f_i}{\epsilon}\right) = \left[\frac{a_i}{\sum_{j=1}^N \exp\left(\frac{g_j - C_{ij}}{\epsilon}\right)}\right]^{\frac{\tau_a}{\tau_a + \epsilon}}$$

Then we optimize $\boldsymbol{g}$ as follows:

$$\frac{\partial \mathcal{J}_\mathrm{E}}{\partial g_j} = -b_j \exp\left(-\frac{g_j}{\tau_b}\right) + \exp\left(\frac{g_j}{\epsilon}\right) \sum_{i=1}^M \exp\left(\frac{f_i - C_{ij}}{\epsilon}\right) = 0,$$

and thus,

$$\exp\left(\frac{g_j}{\epsilon}\right) = \left[\frac{b_j}{\sum_{i=1}^M \exp\left(\frac{f_i - C_{ij}}{\epsilon}\right)}\right]^{\frac{\tau_b}{\tau_b + \epsilon}}$$

In summary, we can rewrite the results via involving two new variables $f_i' = \exp\left(\frac{f_i}{\epsilon}\right)$ and $g_i' = \exp\left(\frac{g_i}{\epsilon}\right)$

$$
\begin{cases}
f_i'^{(t)} = \left[\dfrac{a_i}{\sum_{j=1}^{N} g_j'^{(t-1)} \exp\left(-\frac{C_{ij}}{\epsilon}\right)}\right]^{\frac{\tau_a}{\tau_a+\epsilon}} \\
g_j'^{(t)} = \left[\dfrac{b_j}{\sum_{i=1}^{M} f_i'^{(t)} \exp\left(-\frac{C_{ij}}{\epsilon}\right)}\right]^{\frac{\tau_b}{\tau_b+\epsilon}}
\end{cases},
$$

and,

$$
\pi_{ij}^{(t)} = f_i'^{(t)} g_j'^{(t)} \exp\left(-\frac{C_{ij}}{\epsilon}\right).
$$

### A.2 IMPLEMENTATION DETAILS

The implementation of CoOp (Zhou et al., 2022b) has different versions with different class token positions and parameter initialization strategies. The MUOT-CLIP model is constructed on CoOp (Zhou et al., 2022b), with the position of the class token at the end and the random parameter initialization strategy. The length of learnable context tokens is set as 16. Following previous work (Zhou et al., 2022b; Chen et al., 2023), we adopt ResNet-50 (He et al., 2016) as the backbone of the image encoder. The number of local visual feature is $M = 7 \times 7$. The parameters of UOT are set as $\tau_a = \tau_b = 1, \epsilon = 0.1$. The number of prompts is set as $N = 4$ when comparing to the baselines. We repeat the experiments three times to obtain the average accuracy of each method.

We train MUOT-CLIP via the SGD optimizer with 0.002 initial learning rate, CosineAnnealingLR schedule, and a warmup trick with $10^{-5}$ learning rate. For small datasets such as FGVCAircraft (Maji et al., 2013), Flowers102 (Nilsback & Zisserman, 2008), and StanfordCars (Krause et al., 2013), the batch size is set as 32, while for the larger datasets such as ImageNet (Deng et al., 2009) and SUN397 (Xiao et al., 2010), the batch size is set as 128. All experiments are executed on 4 NVIDIA RTX 4090D GPUs.

The detailed training and testing pipeline are in Algorithm 1 and 2 respectively.

---

**Algorithm 1** Training Pipeline of MUOT-CLIP

---

**Input**: Few-shot labeled image data $\{(x_i^s, y_i^s)\}$, pretrained CLIP model with text encoder $\mathcal{T}$ and image encoder $\mathcal{I}$.
**Parameter**: The value of $\tau_a$, $\tau_b$ and $\epsilon$, maximum number of inner and outer iteration $T_{in}$ and $T_{out}$.
**Output**: Learned prompts $\mathbf{V_k}$.

1: Initialize $\mathbf{V}_k = \{\mathcal{V}_k^{(n)}\}_{n=1}^N$.
2: **for** $t_{out} = 1, 2, 3, \ldots, T_{out}$ **do**
3:      Obtain the visual feature set $\mathbf{F}_i^s$ via $\mathcal{I}(x_i^s)$.
4:      Obtain the textual feature set $\mathbf{G}_k$ via $\{\mathcal{T}(\mathcal{V}_k^{(n)})\}_{n=1}^N$.
5:      Calculate the cost matrix $\mathbf{C_k} = 1 - \frac{\mathbf{F}_i^q \mathbf{G}_k}{\|\mathbf{F}_i^q\|\|\mathbf{G}_k\|}$.
6:      Tackle the EUOT problem via Eq. (12) with $T_{in}$ iterations to obtain $U_i^{(T_{in})}$ and $V^{(T_{in})}$.
7:      Calculate the optimal transport plan $\boldsymbol{\pi}^*$ via Eq. (13).
8:      Calculate the inter-modality distance $d_T(\mathbf{F}_i^s, G_k)$ via Eq. (4).
9:      Calculate the predicted probability $\hat{P}(y_i^s = k|x_i^s, \mathbf{V})$ via Eq. (5).
10:     Update the prompts with $\mathcal{L}_U$ of Eq. (6).
11: **end for**
12: **return** $\mathbf{V}_k^*$

---

### A.3 EXECUTION TIME EXPERIMENTS

To evaluate the efficiency of MUOT-CLIP, we calculate the execution time of PLOT and MUOT-CLIP. Specially, we choose a relatively small dataset EuroSAT and a larger dataset ImageNet to analyze the scalability of the methods. The total execution time on EuroSAT and ImageNet are

---

**Algorithm 2** Inference Pipeline of MUOT-CLIP

---

**Input**: Query images $\{x_i^q\}$, pretrained CLIP model with text encoder $\mathcal{T}$ and image encoder $\mathcal{I}$, Learned prompts $\mathbf{V}_k^*$
**Parameter**: The value of $\tau_a$, $\tau_b$ and $\epsilon$, maximum number of inner and outer iteration $T_{in}$.
**Output**: Predicted class $k^*$.
  1: Obtain the visual feature of query image $\mathbf{F}_i^q$ via $\mathcal{I}(x_i^q)$.
  2: Obtain the average visual feature of retrieved images $\bar{\mathbf{F}}_k^s$ via Eq. (8).
  3: Obtain the textual feature set $\mathbf{G}_k^*$ via text encoder $\mathcal{T}$.
  4: Calculate the inter-modality distance $d_T(\mathbf{F}_i^s, \mathbf{G}_k)$ of Eq. (9) via Eq. (12) with $T_{in}$ iterations.
  5: Calculate the intra-modality distance $d_T(\mathbf{F}_i^q, \bar{\mathbf{F}}_k^s)$ of Eq. (10) via Eq. (12) with $T_{in}$ iterations.
  6: Obtain $d_k$ via Eq. (11).
  7: **return** $k^* = \min_k d_k$

---

reported in Table 5 and Table 6. Due to the scalable UOT solver, the latency of MUOT-CLIP is modest on EuroSAt compared to PLOT. On the larger ImageNet dataset, the execution time of MUOT-CLIP is less than that of PLOT in most cases.

| Shots | 1 | 2 | 4 | 8 | 16 |
|---|---|---|---|---|---|
| PLOT | 28s | 55s | 58s | 141s | 171s |
| MUOT-CLIP | 37s | 68s | 67s | 143s | 175s |

Table 5: Execution time of PLOT and MUOT-CLIP when training on EuroSAT. The random seed is fixed as 1.

| Shots | 1 | 2 | 4 | 8 | 16 |
|---|---|---|---|---|---|
| PLOT | 316s | 519s | 767s | 1733s | 3369s |
| MUOT-CLIP | 288s | 467s | 817s | 1742s | 3365s |

Table 6: Execution time of PLOT and MUOT-CLIP when training on ImageNet. The random seed is fixed as 1.

| Shots | 1 | 2 | 4 | 8 | 16 |
|---|---|---|---|---|---|
| Tip-Adapter-F | 85.70 ± 0.16 | 86.05 ± 0.46 | 86.40 ± 0.29 | 87.66 ± 0.28 | 89.08 ± 0.27 |
| LP++ | 84.24 ± 1.36 | 85.74 ± 0.56 | 86.94 ± 0.48 | 87.71 ± 0.65 | 88.38 ± 0.61 |
| PLOT | 87.27 ± 0.85 | 87.13 ± 0.60 | 88.57 ± 0.11 | 87.17 ± 0.55 | 87.10 ± 0.44 |
| PLOT+"T-I" | 87.70 ± 0.66 | 88.07 ± 0.31 | 88.47 ± 0.38 | 87.83 ± 0.78 | 87.77 ± 0.06 |
| PLOT+"T-T" | 86.77 ± 0.64 | 86.83 ± 0.60 | 88.67 ± 0.25 | 87.13 ± 0.42 | 86.97 ± 0.32 |
| MOUT-CLIP | 87.20 ± 0.53 | 87.73 ± 0.32 | 88.63 ± 0.61 | 87.50 ± 0.53 | 87.70 ± 0.10 |

Table 7: Comparison of the mean and standard deviation of the accuracy on OxfordPets dataset.

| Shots | 1 | 2 | 4 | 8 | 16 |
|---|---|---|---|---|---|
| Tip-Adapter-F | 67.73 ± 0.57 | 68.18 ± 0.84 | 71.17 ± 0.67 | 84.11 ± 0.49 | 93.02 ± 0.28 |
| LP++ | 78.21 ± 1.01 | 84.69 ± 0.70 | 89.56 ± 0.45 | 92.61 ± 0.32 | 94.26 ± 0.24 |
| PLOT | 72.00 ± 0.50 | 82.00 ± 1.31 | 88.27 ± 0.76 | 92.63 ± 0.47 | 95.37 ± 0.68 |
| PLOT+"T-I" | 72.07 ± 0.61 | 81.30 ± 0.46 | 87.80 ± 0.26 | 92.80 ± 0.10 | 95.10 ± 0.10 |
| PLOT+"T-T" | 74.17 ± 0.38 | 82.70 ± 1.32 | 88.60 ± 0.87 | 92.77 ± 0.40 | 95.40 ± 0.44 |
| MOUT-CLIP | 75.30 ± 0.95 | 82.83 ± 0.49 | 88.60 ± 0.20 | 93.03 ± 0.12 | 95.13 ± 0.31 |

Table 8: Comparison of the mean and standard deviation of the accuracy on Flowers102 dataset.

| Shots | 1 | 2 | 4 | 8 | 16 |
|---|---|---|---|---|---|
| Tip-Adapter-F | 18.23 ± 0.19 | 19.12 ± 0.20 | 20.55 ± 0.20 | 23.60 ± 0.29 | 30.37 ± 0.25 |
| LP++ | 19.69 ± 0.39 | 21.58 ± 0.46 | 24.22 ± 0.60 | 27.73 ± 0.48 | 31.73 ± 0.44 |
| PLOT | 17.77 ± 1.25 | 19.37 ± 0.76 | 22.80 ± 1.25 | 26.63 ± 0.71 | 31.30 ± 0.36 |
| PLOT+"T-I" | 19.13 ± 0.42 | 19.73 ± 0.49 | 22.90 ± 0.96 | 27.13 ± 0.68 | 32.03 ± 0.45 |
| PLOT+"T-T" | 19.07 ± 0.78 | 20.50 ± 0.70 | 24.17 ± 1.23 | 27.67 ± 0.70 | 32.60 ± 0.79 |
| MOUT-CLIP | 20.10 ± 0.35 | 21.10 ± 0.44 | 24.93 ± 0.29 | 28.33 ± 0.67 | 33.53 ± 0.29 |

Table 9: Comparison of the mean and standard deviation of the accuracy on FGCVAircraft dataset.

| Shots | 1 | 2 | 4 | 8 | 16 |
|---|---|---|---|---|---|
| Tip-Adapter-F | 46.92 ± 1.01 | 48.50 ± 1.08 | 57.16 ± 0.53 | 62.38 ± 0.47 | 65.23 ± 0.82 |
| LP++ | 46.97 ± 1.37 | 52.44 ± 0.99 | 57.75 ± 0.82 | 62.42 ± 0.53 | 66.40 ± 0.50 |
| PLOT | 47.23 ± 2.23 | 51.17 ± 3.71 | 55.70 ± 0.44 | 61.60 ± 0.66 | 65.20 ± 0.53 |
| PLOT+"T-I" | 47.37 ± 1.39 | 53.30 ± 1.39 | 55.93 ± 0.81 | 62.63 ± 0.68 | 66.20 ± 0.62 |
| PLOT+"T-T" | 47.67 ± 2.90 | 51.90 ± 3.53 | 57.97 ± 0.59 | 62.57 ± 1.10 | 66.10 ± 0.52 |
| MOUT-CLIP | 48.87 ± 2.00 | 54.57 ± 0.64 | 58.27 ± 0.59 | 63.43 ± 0.67 | 66.57 ± 1.00 |

Table 10: Comparison of the mean and standard deviation of the accuracy on DTD dataset.

| Shots | 1 | 2 | 4 | 8 | 16 |
|---|---|---|---|---|---|
| Tip-Adapter-F | 47.63 ± 2.64 | 57.62 ± 1.86 | 69.30 ± 2.41 | 75.22 ± 1.32 | 78.59 ± 1.48 |
| LP++ | 57.23 ± 1.63 | 61.65 ± 1.66 | 68.67 ± 2.21 | 75.86 ± 0.99 | 80.53 ± 1.05 |
| PLOT | 56.20 ± 4.60 | 64.00 ± 4.12 | 70.77 ± 2.12 | 78.00 ± 2.26 | 82.70 ± 0.17 |
| PLOT+"T-I" | 59.23 ± 2.30 | 63.23 ± 2.08 | 72.63 ± 0.70 | 79.70 ± 1.54 | 82.83 ± 0.35 |
| PLOT+"T-T" | 57.70 ± 3.05 | 64.37 ± 0.60 | 72.37 ± 3.52 | 78.50 ± 2.72 | 83.07 ± 0.81 |
| MOUT-CLIP | 62.20 ± 2.04 | 66.17 ± 0.29 | 75.30 ± 1.22 | 80.53 ± 2.61 | 82.93 ± 0.42 |

Table 11: Comparison of the mean and standard deviation of the accuracy on EuroSAT dataset.

| Shots | 1 | 2 | 4 | 8 | 16 |
|---|---|---|---|---|---|
| Tip-Adapter-F | 57.24 ± 0.23 | 58.12 ± 0.50 | 59.34 ± 0.20 | 64.25 ± 0.19 | 71.38 ± 0.23 |
| LP++ | 57.20 ± 0.65 | 59.95 ± 0.36 | 63.44 ± 0.34 | 67.81 ± 0.24 | 72.33 ± 0.18 |
| PLOT | 56.17 ± 0.59 | 58.40 ± 0.66 | 62.70 ± 0.70 | 67.03 ± 0.49 | 72.90 ± 1.21 |
| PLOT+"T-I" | 57.23 ± 0.21 | 58.60 ± 0.89 | 63.03 ± 0.74 | 67.50 ± 0.53 | 72.27 ± 0.31 |
| PLOT+"T-T" | 56.27 ± 0.55 | 58.30 ± 0.50 | 63.00 ± 0.90 | 67.20 ± 0.44 | 73.07 ± 0.95 |
| MOUT-CLIP | 57.63 ± 0.51 | 58.80 ± 0.80 | 63.33 ± 0.70 | 67.47 ± 0.64 | 72.57 ± 0.06 |

Table 12: Comparison of the mean and standard deviation of the accuracy on StanfordCars dataset.

| Shots | 1 | 2 | 4 | 8 | 16 |
|---|---|---|---|---|---|
| Tip-Adapter-F | 77.53 ± 0.14 | 77.53 ± 0.22 | 77.82 ± 0.27 | 78.26 ± 0.22 | 78.99 ± 0.15 |
| LP++ | 76.61 ± 0.77 | 77.22 ± 0.55 | 77.79 ± 0.34 | 78.53 ± 0.14 | 78.88 ± 0.19 |
| PLOT | 78.03 ± 0.06 | 77.90 ± 0.50 | 77.30 ± 0.26 | 75.50 ± 0.17 | 77.17 ± 0.51 |
| PLOT+"T-I" | 78.00 ± 0.17 | 78.33 ± 0.21 | 78.03 ± 0.25 | 76.13 ± 0.25 | 77.30 ± 0.17 |
| PLOT+"T-T" | 77.70 ± 0.10 | 77.80 ± 0.53 | 77.27 ± 0.31 | 75.63 ± 0.23 | 77.23 ± 0.35 |
| MOUT-CLIP | 77.97 ± 0.15 | 78.13 ± 0.31 | 77.93 ± 0.25 | 76.13 ± 0.15 | 77.47 ± 0.12 |

Table 13: Comparison of the mean and standard deviation of the accuracy on Food101 dataset.

| Shots | 1 | 2 | 4 | 8 | 16 |
|---|---|---|---|---|---|
| Tip-Adapter-F | 61.02 ± 0.36 | 62.15 ± 0.28 | 63.86 ± 0.19 | 67.25 ± 0.16 | 70.94 ± 0.13 |
| LP++ | 62.47 ± 0.27 | 64.65 ± 0.25 | 67.28 ± 0.27 | 69.34 ± 0.14 | 71.23 ± 0.07 |
| PLOT | 62.63 ± 0.50 | 62.37 ± 0.35 | 65.27 ± 0.45 | 66.43 ± 0.67 | 69.63 ± 0.20 |
| PLOT+"T-I" | 62.63 ± 0.32 | 62.97 ± 0.49 | 65.27 ± 0.67 | 66.77 ± 0.15 | 69.60 ± 0.10 |
| PLOT+"T-T" | 63.20 ± 0.56 | 62.77 ± 0.74 | 65.80 ± 0.56 | 67.33 ± 0.49 | 70.37 ± 0.15 |
| MOUT-CLIP | 63.43 ± 0.46 | 63.77 ± 0.23 | 66.20 ± 0.53 | 67.80 ± 0.17 | 70.30 ± 0.10 |

Table 14: Comparison of the mean and standard deviation of the accuracy on SUN397 dataset.

| Shots | 1 | 2 | 4 | 8 | 16 |
|---|---|---|---|---|---|
| Tip-Adapter-F | 87.35 ± 0.64 | 88.17 ± 0.29 | 89.49 ± 0.25 | 90.54 ± 0.34 | 92.10 ± 0.25 |
| LP++ | 88.56 ± 0.43 | 89.53 ± 0.35 | 90.87 ± 0.19 | 91.84 ± 0.24 | 92.73 ± 0.17 |
| PLOT | 89.03 ± 0.25 | 89.87 ± 0.25 | 90.67 ± 0.06 | 90.93 ± 0.45 | 92.93 ± 0.60 |
| PLOT+"T-I" | 89.23 ± 0.12 | 89.73 ± 0.15 | 91.03 ± 0.06 | 91.50 ± 0.20 | 93.03 ± 0.49 |
| PLOT+"T-T" | 89.53 ± 0.15 | 90.13 ± 0.35 | 91.33 ± 0.12 | 91.10 ± 0.46 | 93.10 ± 0.62 |
| MOUT-CLIP | 89.67 ± 0.15 | 90.67 ± 0.29 | 91.73 ± 0.15 | 91.93 ± 0.32 | 93.37 ± 0.38 |

Table 15: Comparison of the mean and standard deviation of the accuracy on Caltech101 dataset.

| Shots | 1 | 2 | 4 | 8 | 16 |
|---|---|---|---|---|---|
| Tip-Adapter-F | 64.28 ± 0.54 | 65.48 ± 0.43 | 67.61 ± 0.28 | 72.05 ± 0.53 | 77.30 ± 0.21 |
| LP++ | 65.41 ± 0.37 | 69.20 ± 0.52 | 71.68 ± 0.41 | 74.86 ± 0.36 | 77.46 ± 0.39 |
| PLOT | 64.37 ± 0.48 | 67.83 ± 0.05 | 70.67 ± 1.09 | 75.80 ± 0.52 | 77.77 ± 0.38 |
| PLOT+"T-I" | 64.57 ± 0.15 | 67.70 ± 0.26 | 71.77 ± 0.29 | 77.07 ± 1.16 | 78.77 ± 0.35 |
| PLOT+"T-T" | 65.60 ± 0.95 | 69.27 ± 0.40 | 72.37 ± 0.87 | 75.93 ± 0.40 | 78.07 ± 0.15 |
| MOUT-CLIP | 66.17 ± 0.55 | 69.87 ± 0.25 | 73.23 ± 0.12 | 77.30 ± 0.79 | 78.83 ± 0.25 |

Table 16: Comparison of the mean and standard deviation of the accuracy on UCF101 dataset.

| Shots | 1 | 2 | 4 | 8 | 16 |
|---|---|---|---|---|---|
| Tip-Adapter-F | 60.59 ± 0.14 | 61.42 ± 0.05 | 62.12 ± 0.06 | 63.41 ± 0.07 | 65.06 ± 0.04 |
| LP++ | 61.18 ± 0.08 | 61.56 ± 0.14 | 62.55 ± 0.12 | 63.76 ± 0.07 | 64.73 ± 0.05 |
| PLOT | 57.90 ± 1.56 | 59.90 ± 0.50 | 60.37 ± 1.31 | 60.57 ± 0.93 | 60.67 ± 0.38 |
| PLOT+"T-I" | 59.53 ± 0.25 | 59.83 ± 0.21 | 60.03 ± 0.38 | 60.53 ± 0.75 | 60.30 ± 0.35 |
| PLOT+"T-T" | 58.20 ± 1.32 | 60.13 ± 0.50 | 60.77 ± 0.81 | 61.33 ± 0.72 | 61.93 ± 0.35 |
| MOUT-CLIP | 59.63 ± 0.31 | 60.20 ± 0.17 | 60.77 ± 0.35 | 61.70 ± 0.53 | 62.23 ± 0.21 |

Table 17: Comparison of the mean and standard deviation of the accuracy on ImageNet dataset.

## A.4 DETAILED RESULTS FOR COMPARISON AND ABLATION

For each method, we run 3 times under each setting. The mean and standard deviation of the accuracy of Tip-Adapter-F (Zhang et al., 2022), LP++ (Huang et al., 2024), PLOT (Chen et al., 2023) , MUOT-CLIP and its variants on all the 11 datasets are reported in Table 7-17. MUOT-CLIP consistently achieves competitive performance over the baselines. In addition, the overall performance of the ablated versions of MUOT-CLIP are better than that of PLOT (Chen et al., 2023), which proves the effectiveness of UOT based inter- and intra- modality distance measure as well as the contributions of each components.

| Shots | 1 | 2 | 4 | 8 | 16 | 32 | 64 |
|---|---|---|---|---|---|---|---|
| OxfordPets | 87.2 | 87.73 | 88.63 | 87.5 | 87.7 | 88.7 | **89.87** |
| Flowers102 | 75.3 | 82.83 | 88.6 | 93.03 | 95.13 | 96.63 | **96.97** |
| FGCVAircraft | 20.1 | 21.1 | 24.93 | 28.33 | 33.53 | **36.4** | 35.77 |
| DTD | 48.87 | 54.57 | 58.27 | 63.43 | 66.57 | 69.33 | **71.83** |
| EuroSAT | 62.20 | 66.17 | 75.30 | 80.53 | 82.93 | 85.87 | **88.3** |
| StanfordCars | 57.63 | 58.8 | 63.33 | 67.47 | 72.57 | 75.53 | **75.63** |
| Food101 | 77.97 | 78.13 | 77.93 | 76.13 | 77.47 | 78.4 | **79.33** |
| SUN397 | 63.43 | 63.77 | 66.2 | 67.8 | 70.3 | 71.93 | **72.13** |
| Caltech101 | 89.67 | 90.67 | 91.73 | 91.93 | 93.37 | 93.77 | **93.9** |
| UCF101 | 66.17 | 69.87 | 73.23 | 77.30 | 78.83 | 80.23 | **81.30** |
| Euro. | 62.20 | 66.17 | 75.30 | 80.53 | 82.93 | 85.87 | **88.30** |
| ImgN. | 59.63 | 60.20 | 60.77 | 61.70 | 62.23 | 62.43 | **63.00** |

Table 18: Detailed Results for Shots Ablation.

| $N$ | 1 | 2 | 4 | 8 |
|---|---|---|---|---|
| OxfordPets | 87.33 ± 0.90 | 87.87 ± 0.21 | 88.63 ± 0.61 | **88.70** ± 0.26 |
| Flowers102 | 87.30 ± 0.62 | 88.27 ± 0.42 | 88.60 ± 0.20 | **88.70** ± 0.10 |
| FGCVAircraft | 23.90 ± 2.67 | 23.90 ± 1.67 | **24.93** ± 0.29 | 24.80 ± 0.61 |
| DTD | 57.77 ± 0.40 | 58.10 ± 1.01 | 58.27 ± 0.59 | **58.33** ± 0.35 |
| EuroSAT | 74.20 ± 0.44 | 75.20 ± 1.15 | **75.30** ± 1.22 | 75.17 ± 0.60 |
| StanfordCars | 62.30 ± 0.52 | 62.97 ± 1.11 | **63.33** ± 0.70 | 63.27 ± 0.29 |
| Food101 | 76.80 ± 1.37 | 77.03 ± 1.05 | 77.93 ± 0.25 | **78.43** ± 0.21 |
| SUN397 | 66.13 ± 0.25 | 65.93 ± 0.35 | 66.20 ± 0.53 | **66.93** ± 0.32 |
| Caltech101 | 90.83 ± 0.38 | 91.33 ± 0.32 | 91.73 ± 0.15 | **91.73** ± 0.38 |
| UCF101 | 71.57 ± 0.40 | 72.43 ± 0.25 | **73.23** ± 0.12 | 73.10 ± 0.17 |

Table 19: Prompts number ablation on different datasets. The number of shots is fixed as 4.

## A.5 DETAILED RESULTS FOR SHOTS ABLATION

To evaluate the performance of MUOT-CLIP with more shots (e.g., 32 shots and 64 shots), we conduct extensive experiments on all the 11 datasets with 1, 2, 4, 8, 16, 32, and 64 shots respectively. The number of prompts is fixed as $N = 4$. In almost all the datasets, MUOT-CLIP achieves the highest accuracy with 64 shots, which proves its ability to be scalable to more labeled training data. The detailed results are in Table 18.

## A.6 DETAILED RESULTS FOR PROMPTS NUMBER ABLATION

We evaluate the performance of MUOT-CLIP with different number of prompts on various datasets. The results are reported in Table 19.

