# OpenReview forum: "MUOT-CLIP: Enhancing Few-Shot Adaptation of CLIP via Inter- and Intra- Modality Unbalanced Optimal Transport"
_ICLR.cc/2026/Conference — ICLR 2026 Conference Withdrawn Submission_

### Official Review · Reviewer_eVu4 · 2025-10-28

**Soundness:** 3
**Presentation:** 4
**Contribution:** 3
**Rating:** 6
**Confidence:** 5

**Summary:**

This paper proposes MUOT-CLIP, a novel prompt learning framework that enhances the few-shot adaptation capability of CLIP by formulating both inter-modality (text-image) and intra-modality (image-image) distance measures using Unbalanced Optimal Transport (UOT). The authors argue that classical OT, as used in prior work like PLOT, is suboptimal for this task as it cannot filter out noisy or misleading features. MUOT-CLIP introduces a three-step process: UOT-guided prompt learning, prototype image retrieval, and a combined UOT-based inference, and demonstrates state-of-the-art performance on 11 few-shot classification benchmarks.

**Strengths:**

The core idea of replacing Classical OT with UOT for measuring feature distances in few-shot learning is well-motivated and novel. The argument that UOT can adaptively filter noise by relaxing marginal constraints is convincing and addresses a clear weakness in the PLOT method.

The paper provides extensive experiments across 11 diverse datasets and multiple shot settings (1 to 16). The results are compelling, showing consistent and often significant improvements over strong baselines, including adapter-based (Tip-Adapter, CLIP-Adapter), prompt-based (CoOp, PLOT), and linear probing (LP++) methods.

The combination of inter- and intra-modality UOT during inference is a clever design to mitigate the modality gap issue. The ablation studies (Table 2) convincingly show that both components contribute positively to the final performance.

**Weaknesses:**

Comparison to Broader Baselines: While the selected baselines are well-established, the field of efficient adaptation for vision-language models has progressed rapidly. To more firmly position the contribution of MUOT-CLIP within the current state-of-the-art, I would recommend including comparisons with more recent works from 2024 (and even 2025, if available). It would provide a more up-to-date and compelling performance benchmark. In addition, the comparison is comprehensive within the specific line of CLIP adaptation, but it could be slightly broader. For instance, a comparison with a simple fine-tuning baseline is missing.

Clarity on Retrieval Mechanism: The Memory and Retrieval (M&R) step for obtaining prototype images is described at a high level (e.g., "full retrieval or d(F_i^s, G_k^s) guided partial retrieval"). The specific implementation used for the main results and its impact on performance could be described more precisely.

**Questions:**

In the M&R step for intra-modality distance, what was the exact retrieval strategy (full or partial) used for the main results in Table 1? Was the performance sensitive to this choice, and did you experiment with more sophisticated retrieval mechanisms?

The parameter μ in Eq. (8), which balances the inter- and intra-modality distances, is crucial. How was this parameter set in your experiments? Was it tuned per dataset/shot, or was a fixed value used? An ablation on its sensitivity would be informative.

---

### Official Review · Reviewer_r8Ja · 2025-10-29

**Soundness:** 3
**Presentation:** 4
**Contribution:** 2
**Rating:** 4
**Confidence:** 3

**Summary:**

The paper proposes MUOT-CLIP, a few-shot adaptation framework for CLIP that introduces Unbalanced Optimal Transport (UOT) to measure both inter-modality and intra-modality distances. Compared with the classical OT, MUOT-CLIP relaxes the equality constraints of OT to adaptively filter such noise and mitigate the modality gap between visual and textual embeddings. The authors further develop a scalable Sinkhorn-like UOT solver with entropy regularization for efficiency. Experiments on 11 standard datasets demonstrate the effectiveness of the proposed MUOT-CLIP.

**Strengths:**

1. The motivation is clear; the authors identify meaningful shortcomings of classical OT and modality gap in CLIP adaptation.
2. The paper is clearly written overall with good structure, which is easy to follow.
3. The paper provides detailed and convincing theoretical analysis for the proposed UOT formulation. The derivations of the optimization process and the scalable Sinkhorn-like solver are well presented.

**Weaknesses:**

1. The related work section provides only a brief overview of Optimal Transport and lacks sufficient discussion. A deeper review would strengthen the paper’s technical grounding and contextual relevance.
2. The experiments are only conducted on CLIP-ResNet-50, which restricts the generality of the conclusions. Results on other common backbones such as CLIP-ViT are missing, as well as evaluations on benchmarks with distribution shift, like ImageNet-A/V/K/S. Including these results would provide stronger evidence for the robustness and scalability of the proposed method.
3. Most of the baselines are from 2022–2024, while several recent few-shot or prompt-learning approaches are not included. This limits the credibility of the claimed SOTA performance. The authors should update the comparisons to reflect the current landscape of CLIP adaptation methods.
4. Replacing classical OT with unbalanced OT is a relatively straightforward extension rather than a fundamentally new framework. The architectural design largely follows existing work, with only minor modifications in the transport formulation. Therefore, the authors need to carefully demonstrate the novelty of the proposed method.

**Questions:**

please refer to Weaknesses.

---

### Official Review · Reviewer_oj6h · 2025-11-01

**Soundness:** 1
**Presentation:** 2
**Contribution:** 1
**Rating:** 2
**Confidence:** 4

**Summary:**

This paper proposes MUOT-CLIP, a novel prompt-based optimal transport (OT) method for few-shot classification in VLMs. Unlike classical OT, the method relaxes transport constraints via UOT to mitigate noisy distance measurements. Additionally, the authors introduce inter- and intra-modality UOT-based inference. Experiments on few-shot classification benchmarks confirm its effectiveness.

**Strengths:**

-	The written and figures are clear.

**Weaknesses:**

-	The motivation is unclear. The reason for using Optimal Transport (OT) to address few-shot adaptation under CLIP is not well justified.
-	There are significant issues with the reported results. The Tip-Adapter-F performance in Table 1 does not match the original paper’s findings (e.g., the original reports 75.81 average accuracy for 16-shot, while this paper’s results diverge).
-	The experiments are insufficient:
    - Only few-shot classification results are provided. Additional evaluations (e.g., OOD generalization, base-to-new generalization, or results across different architectures) are needed to demonstrate the method’s effectiveness.
    - The baselines are outdated (2022–2023 works). Comparisons with recent state-of-the-art methods are lacking. Notably, some training-free approaches [1,2] even outperform this work in few-shot classification.
    - Need additional ablation of UOT and OT.
-	Training time and GPU memory usage comparisons between MUOT-CLIP and other baselines should be included.
-	The proposed approach appears to combine the drawbacks of both prompt-based and cache-based methods, resulting in slow training and additional memory consumption.

[1] ProKeR: A Kernel Perspective on Few-Shot Adaptation of Large Vision-Language Models. In CVPR 25.
[2] A hard-to-beat baseline for training-free clip-based adaptation. In ICLR 24.

**Questions:**

See above.

---

### Official Review · Reviewer_xR3t · 2025-11-07

**Soundness:** 3
**Presentation:** 3
**Contribution:** 2
**Rating:** 4
**Confidence:** 4

**Summary:**

See Questions

**Strengths:**

See Questions

**Weaknesses:**

See Questions

**Questions:**

After reading the paper, I have the following comments and suggestions. I hope the authors could address them carefully.

- Q1. The abstract is overly long in its background and motivation, while the technical description of the proposed method is minimal—only about three lines. A more balanced structure is recommended, with a clearer summary of the core contributions.

- Q2. The motivation presented in the abstract—“classical OT, which forces equality constraints on both the source and target weights of the transport plan, is susceptible to noises”—is not clearly explained. Additionally, the illustrative example mentioned should be accompanied by a corresponding figure in the main text to aid reader understanding.

- Q3. CLIP was introduced in 2021, and CLIP-based few-shot learning has since become a very active area of research. However, the related work section mainly discusses well-known methods such as CoOp, CoCoOp, and TIP-Adapter, which is insufficient. Many recent and strongly relevant works are missing from the discussion. For example, include but not limited to: SgVA-CLIP [1], Amu-tuning [2], TIMO [3].
A more comprehensive and up-to-date literature review is necessary.

- Q4. The paper does not clearly explain how Figure 1 is generated. More details about the setup and methodology behind the figure are needed.

- Q5. I am curious about how the proposed MUOT method is implemented in code. The paper would benefit from a more transparent explanation of its computational aspects or implementation details.

- Q6. Minor issues:

(a) There are extra spaces before punctuation marks (e.g., Line 151).

(b) Some symbols and abbreviations (e.g., i.e.) are not properly italicized.


Overall, the paper is well presented. I will consider adjusting my score based on the authors' rebuttal and how well they address the concerns above.

-------

[1] Sgva-clip: Semantic-guided visual adapting of vision-language models for few-shot image classification. TMM 2023.

[2] Amu-tuning: Effective logit bias for clip-based few-shot learning. CVPR 2024

[3] TIMO: Text and Image Are Mutually Beneficial: Enhancing Training-Free Few-Shot Classification with CLIP. AAAI 2025

**Details Of Ethics Concerns:**

See Questions

---

### Note · Authors · 2025-11-13

I have read and agree with the venue's withdrawal policy on behalf of myself and my co-authors.